# Reduction of Potential-Induced-Degradation of p-Type PERC Solar Cell Modules by an Ion-Diffusion Barrier Layer Underneath the Front Glass

**Eunjin Jang** [1], **Kyoung-suk Oh** [2,3] **and Sangwoo Ryu** [1,*]

1   Department of Advanced Materials Engineering, Kyonggi University, Suwon 16227, Korea; eunjin23@kyonggi.ac.kr
2   New & Renewable Energy Research Center, Korea Electronics Technology Institute, Seongnam 13509, Korea; ks.oh@hyundai-es.co.kr
3   PV Module Development Team, Hyundai Energy Solution, Chungcheongbuk-do 27711, Korea
*   Correspondence: sryu@kyonggi.ac.kr; Tel.: +82-31-249-9761

**Abstract:** With the maturation of silicon-based technologies, silicon solar cells have achieved a high conversion efficiency that approaches the theoretical limit. Currently, great efforts are being made to enhance the reliability of silicon solar cells. When the silicon solar cells are made into modules, potential-induced-degradation (PID) occurs during operation because of the high voltage applied between the frame and the cells, which reduces the efficiency and output power. The diffusion of $Na^+$ ions from the front glass and the increased leakage current along the migration path are the major causes of PID. In this work, atomic layer deposition (ALD)-grown amorphous thin $Al_2O_3$ layers are introduced underneath the front glass to prevent the diffusion of $Na^+$ ions and the resulting PID. Accelerated PID tests showed that an ALD-grown $Al_2O_3$ layer of 30 nm could effectively suppress PID seriously affecting the conversion efficiency or light transmittance. The introduction of an ion-diffusion barrier underneath the front glass is expected to contribute to securing the long-term reliability of silicon-based electricity generation, together with the introduction of barrier layers inside the solar cells.

**Keywords:** potential-induced degradation; PERC solar cell module; ion-diffusion barrier; $Al_2O_3$

## 1. Introduction

As the demand for energy continues to increase worldwide, interest in eco-friendly renewable energy has accelerated because of the increasingly serious environmental pollution and global warming, which is recognized as the main culprit of climate change. Solar energy conversion devices, especially photovoltaic cells, have been regarded as most promising among the various types of renewable energy because of their unlimited reserves and high energy conversion efficiency [1,2]. Among the photovoltaic devices, crystalline silicon solar cells have developed most rapidly, because of their high conversion efficiency and a mature silicon-based industry; as a result, they account for more than 80% of the global solar cell market [3–6]. According to the Best Research-Cell Efficiency chart reported by the National Renewable Energy Laboratory, the highest cell efficiency for a single-crystalline silicon solar cell is 26.1%, achieved by the Institute for Solar Energy Research, Germany [7].

Most solar cell modules provided in the market guarantee operation for longer than 25 years. As the conversion efficiency approaches the theoretical limit of 29%, the interest is shifting from efficiency to long-term stability.

When silicon solar cell modules are connected in series to produce a high voltage of 600–1000 V, the same high voltage is applied between the grounded module frame and the solar cell, which causes a large leakage current and loss of power output during the production of electricity. This is referred to as potential-induced-degradation (PID), which

deteriorates the photovoltaic performance of crystalline silicon solar cell modules over a long period of time [8–12].

PID is known to occur as a result of the diffusion of $Na^+$ ions present in the soda-lime front glass that is generally used for silicon solar cell modules and contains 12% $Na_2O$. When a high voltage is applied, $Na^+$ ions inside the front glass diffuse into the cells across the encapsulant and anti-reflection coatings. The diffused $Na^+$ ions accumulate at the interface between the anti-reflection coatings and the cells or at the stacking faults inside the silicon, which forms a conduction path, leading to a large leakage current [13–17], which gradually degrades the conversion efficiency and maximum power output of the modules.

The PID can be reduced by adjusting the refractive index of the anti-reflection coatings formed on top of the cells. As the refractive index of silicon nitride ($SiN_x$), which is typically used for the anti-reflection coating of silicon solar cells, increases, the conductivity also increases, resulting in a reduction in the migration of positively charged ions to $SiN_x$. This effectively enhances the resistance to PID. However, if the refractive index becomes higher than 2.14, light with a short wavelength cannot be absorbed by silicon, causing a decrease in the cell efficiency as a result of the reduction in the photocurrent [16–18].

An alternative way to prevent PID is to insert a layer that can block the penetration of ions from the glass [19,20]. This layer can be located between the front glass and the front encapsulant or between the anti-reflection coating and the cell. These blocking layers are known to effectively suppress the diffusion of $Na^+$ ions inside the glass and the shunting path and consequently reduce PID. Possible candidates for this blocking layer include $Al_2O_3$, $TiO_2$, and $SiO_2$ [21–23]. Previous work done by Jung et al. shows that an ultrathin layer of $SiO_x$ formed between the cell and the anti-reflection coating effectively reduces PID [24].

Herein, we explored the effect of $Al_2O_3$ thin layers, which are known to have high light transmittance with outstanding barrier performance against ion diffusion, on the reduction of PID in silicon solar cell modules. As the thickness of $Al_2O_3$ increased, the PID was remarkably reduced, and a 30 nm layer of $Al_2O_3$ could suppress PID effectively.

## 2. Materials and Methods

Ions can diffuse through the grain boundaries in crystalline $Al_2O_3$ layers, and an amorphous phase of $Al_2O_3$ is required for suitable performance as a diffusion barrier. For this purpose, thermal atomic layer deposition (ALD) was utilized for the growth of the $Al_2O_3$ ion diffusion barrier layers. Soda-lime glass and quartz were cut to a size of 5 cm × 5 cm. These substrates were first ultrasonically cleaned with acetone, methanol, and isopropyl alcohol sequentially for 15 min each and finally rinsed with deionized water. The cleaned glass substrates were then dried in a thermal ALD chamber at 150 °C for 30 min. Thermal ALD of $Al_2O_3$ was performed at the same temperature with trimethylaluminum (TMA, $Al(CH_3)_3$) and $H_2O$ as the precursor and reactant, respectively. During each cycle, the substrate was exposed to the precursor for 0.2 s, followed by purging with 80 sccm of Ar for 20 s.

Structural characterization was performed using X-ray diffraction (Empyrean, Malvern Panalytical, Malvern, UK) and scanning electron microscopy (JSM-7610F Plus, Jeol, Tokyo, Japan). The light transmittance was measured using UV-vis spectroscopy (Optizen POP, K LAB, Daejeon, Korea) for 10 to 30 nm $Al_2O_3$ films deposited on quartz substrates.

To characterize the PID, an acceleration test was performed using IEC62804-1, an internationally recognized standard-based solar cell test device [25]. For silicon solar cells, passivated emitter and rear contact (PERC) cells, which represent the largest portion of the market, were used. As shown in Figure 1, p-type PERC solar cells, encapsulants, and $Al_2O_3$-grown sola-lime glass were placed on an aluminum chuck heated to 60 °C on a hot plate [26]. Then, a 2-kg weight was loaded on top of the glass and +1000 V was applied to the bottom aluminum chuck for 96 h. Light I-V curves and electroluminescence (EL) images were obtained to evaluate the effect of the blocking layers on PID. To ensure testing uniformity, four module-like layer stacks underwent an accelerated test at the same time.

Note that PERC solar cells without oxidation layers inserted between the anti-reflection coating and the cell were examined because these oxidation layers inside the cells are known to prevent PID as well.

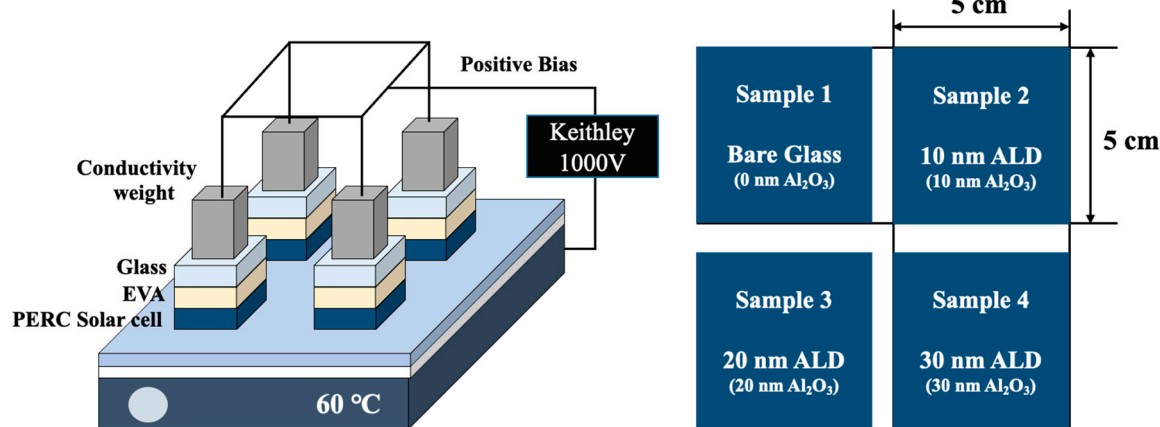

**Figure 1.** Schematic of the accelerated PID test. A 2 kg weight made of aluminum was used. Four module-like stack samples were tested at the same time. After 96 h, each stack was disassembled, and I-V measurements were performed.

## 3. Results and Discussion

### 3.1. Characterization of the ALD-Grown $Al_2O_3$ for Si-Solar Cell Usage

#### 3.1.1. Uniform Growth of $Al_2O_3$ with Amorphous Structure

Figure 2 shows the SEM plan view and cross-section view images of the ALD-grown $Al_2O_3$ thin films on Si wafer (a) and glass (b), (c). For both $Al_2O_3$ films, the calibrated growth rate was 1.3 Å/cycle. SEM plan view images of the $Al_2O_3$ films, as shown in Figure 2a,b, allow reasonable doubt regarding the polycrystalline microstructure. However, all the ALD-grown $Al_2O_3$ thin films investigated in this work exhibited an amorphous structure, which is confirmed by the absence of a pronounced XRD peak (not shown), as is generally expected for films grown by ALD.

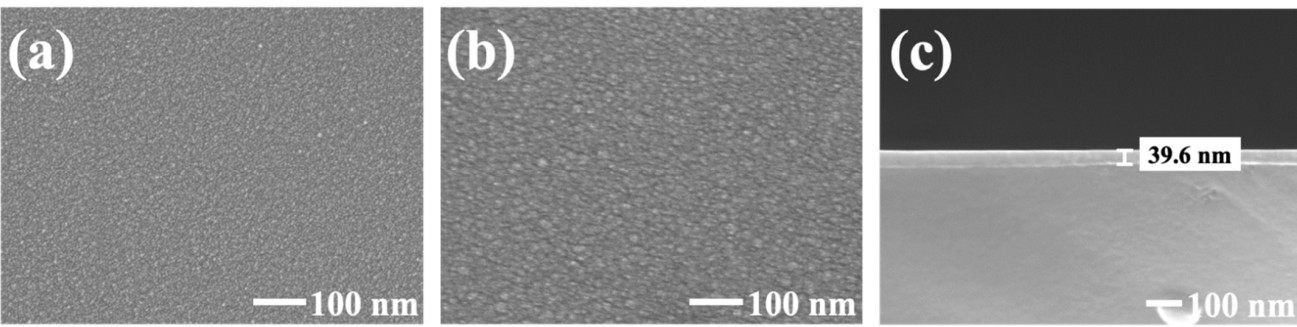

**Figure 2.** SEM plan view images of the ALD-grown $Al_2O_3$ thin films on (**a**) single crystalline Si wafer, (**b**) glass. (**c**) SEM cross-section view image of the ALD-grown $Al_2O_3$ thin films on a glass.

#### 3.1.2. Optical Transmittance and Passivation Properties of $Al_2O_3$ Thin Films

Prior to the accelerated PID test, the light transmittance of $Al_2O_3$-coated quartz substrates with various $Al_2O_3$ thicknesses was characterized, as shown in Figure 3a,b. Note that light transmittance through the glass and light absorption by silicon decreased with a thick $Al_2O_3$ layer. As presented in Figure 3a, the light transmittance in the wavelength range of 200–300 nm was slightly reduced with increasing thickness of $Al_2O_3$. However, the average transmittance for the 200–1000 nm range was greater than 94% and independent

of the $Al_2O_3$ thickness, although the transmittance for 30 nm was slightly smaller than the others.

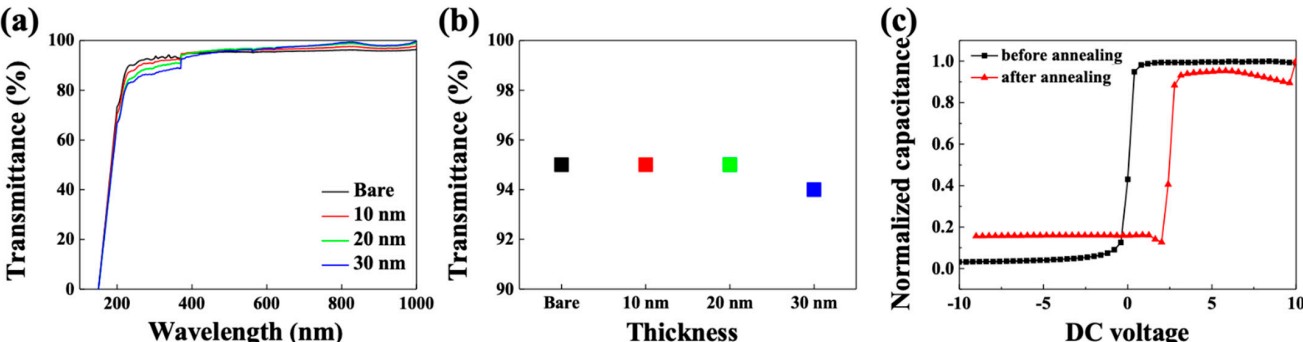

**Figure 3.** (**a**) Light transmittance for 200 to 1000 nm range of ALD-grown $Al_2O_3$ thin films with various thicknesses. (**b**) Average light transmittance with respect to the thickness of the $Al_2O_3$ layer. (**c**) Capacitance-voltage profile of a p-type single-crystalline silicon wafer passivated by the ALD-grown 10 nm $Al_2O_3$ thin layers.

To further confirm the validity of the ALD process used in this work for $Al_2O_3$ thin film growth, we examined the surface passivation properties using capacitance-voltage measurements. For 10 nm-thick $Al_2O_3$ films grown on p-type single-crystalline Si wafers, the capacitance-voltage profile shifted to a positive bias after annealing, as shown in Figure 3I, which is indicative of negatively charged $Al_2O_3$ thin films with the appropriate passivation properties of well-grown $Al_2O_3$ thin films.

### 3.2. Accelerated PID Test with Light I-V and Electroluminescent Measurement

The results of the accelerated PID tests are shown in Figure 4. As mentioned in the Materials and Methods section, the current and voltage with and without light were measured before and after loading a 2 kg weight for 96 h. Note that the reference sample is the initial module before the accelerated PID test. The rest of the samples correspond to the measurement results after the accelerated PID test with respect to the designated $Al_2O_3$ layer thickness. Before the PID test, all the modules showed similar I-V characteristics; therefore, only the module without the $Al_2O_3$ layer was used as the reference.

As shown in Figure 4a, the $Al_2O_3$ layer deposited underneath the front glass can effectively prevent PID. Without the $Al_2O_3$ diffusion barrier, $V_{oc}$ was remarkably reduced, from 0.658 V to 0.572 V. However, a 10 nm-thick layer of $Al_2O_3$ recovers $V_{oc}$ to that of the reference module. $J_{sc}$ does not change significantly, which can be inferred from the similar light absorption properties presented in Figure 3. All the modules exhibited similar series resistance, which can be obtained from the slope that crosses the voltage axis, regardless of the presence of the $Al_2O_3$ diffusion barrier layer. However, slopes that cross the current axis showed different behavior; the module without the $Al_2O_3$ layer showed a quite stiff slope, whereas the slopes of the modules with the $Al_2O_3$ layer recovered close to that of the reference. This is predictable because the leakage current that causes PID is related to the shunt resistance and not to the series resistance.

Although the $V_{oc}$ and $J_{sc}$ of the cells with $Al_2O_3$ layers were almost the same, regardless of the thickness of $Al_2O_3$, the change in the fill factor and the solar conversion efficiency of each module were remarkably reduced in cells with an $Al_2O_3$ layer, as shown in Figure 4b. The fill factor decreased from 75.3% to 72.4% without the $Al_2O_3$ layer, while it recovered to its initial value when employing an $Al_2O_3$ layer of only 10 nm. Without the $Al_2O_3$ layer, the conversion efficiency deteriorated significantly; it dropped by approximately 18.8% after the PID test. However, as the thickness of $Al_2O_3$ increased, the efficiency drop tended to decrease, even though there was some fluctuation. Even with only 10 nm of $Al_2O_3$, the efficiency after the PID test for 96 h was 19.92%, which indicates that the efficiency

decreased by approximately 2.54%. When the thickness of $Al_2O_3$ was 30 nm, the efficiency loss was only 0.98% after the PID test.

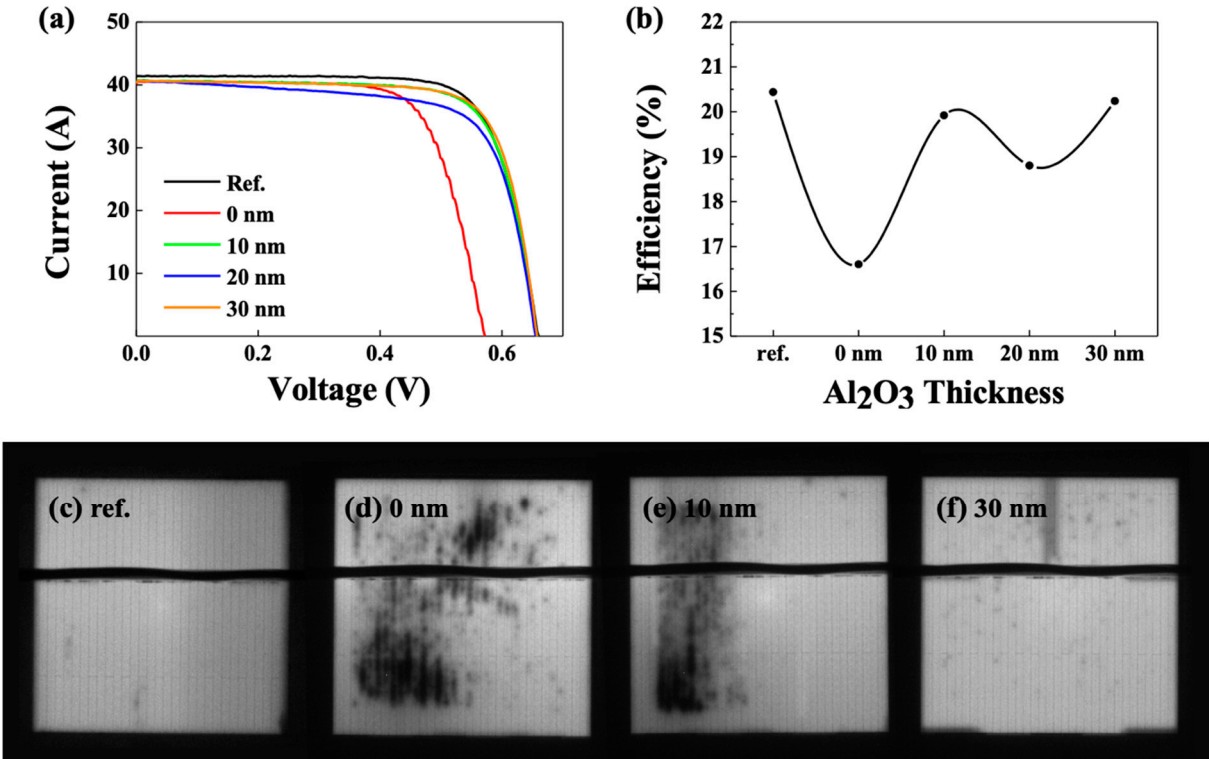

**Figure 4.** Characterization results of PERC cells after the accelerated PID test. (**a**) Light I–V, (**b**) efficiency changes, electroluminescent images obtained after the PID test; (**c**) ref—before the PID test, (**d**) without the $Al_2O_3$ layer, (**e**) 10 nm $Al_2O_3$, and (**f**) 30 nm $Al_2O_3$.

PID causes the formation of defects inside the cell, which can be analyzed by EL. As presented in Figure 4c–f, lots of defects were generated, possibly due to the diffusion of $Na^+$ ions, after the PID test when no $Al_2O_3$ layer was inserted. Although 10 nm-thick $Al_2O_3$ demonstrated a quite reduced PID as shown in light I-V measurement results, EL images of 10 nm indicated that there were still non-negligible number of defects. It could be confirmed that the defects caused by the diffusion of $Na^+$ ions disappeared only when the thickness was 30 nm.

From the light I-V measurement and EL results, it can be concluded that the optimum thickness of $Al_2O_3$, the diffusion barrier of $Na^+$ ions from the front glass, is 30 nm. All the changes of the photovoltaic parameters with various $Al_2O_3$ thicknesses after the accelerated PID test were summarized in Table 1.

**Table 1.** Photovoltaic parameters with various $Al_2O_3$ thicknesses after the accelerated PID test. The changes with respect to the values of reference sample are shown together in the parentheses.

| Sample | $V_{OC}$ (V) | $J_{SC}$ (mA/cm$^2$) | FF (%) | Efficiency (%) |
|---|---|---|---|---|
| Reference | 0.658 | 41.24 | 75.3 | 20.44 |
| 0 nm $Al_2O_3$ | 0.572 (−13.07%) | 40.08 (−2.81%) | 72.4 (−3.85%) | 16.6 (−18.79%) |
| 10 nm $Al_2O_3$ | 0.658 (0%) | 40.32 (−2.23%) | 75.1 (−0.27%) | 19.92 (−2.54%) |
| 20 nm $Al_2O_3$ | 0.655 (−0.46%) | 40.48 (−1.84%) | 70.9 (−5.84%) | 18.8 (−8.02%) |
| 30 nm $Al_2O_3$ | 0.658 (0%) | 40.28 (−2.32%) | 76.4 (1.46%) | 20.24 (−0.98%) |

According to V. Naumann et al., an accumulation of alkali metals should be found at the interface of the front side coatings of the solar cell, which can be demonstrated by lock-in thermography (LIT), and time-of-flight secondary ion mass spectroscopy (ToF-SIMS), when PID occurs [27]. Further investigation using these analysis techniques would manifest the correlation between the occurrence of PID and the diffusion of Na$^+$ ions.

## 4. Conclusions

The improvement in solar cell efficiency has received much recent attention, and intensive research to achieve higher efficiency has been conducted. However, with the widespread use of solar cells, long-term reliability has become another important aspect. PID is regarded as a major source of deterioration that occurs during operation when silicon-based solar cells are made into modules. The diffusion of Na$^+$ ions into the cells and the resulting increase in leakage current are one of the main causes of PID. When ALD-grown amorphous Al$_2$O$_3$ thin layers of 30 nm are introduced underneath the front glass of the module, PID is effectively prevented without a serious decrease in the V$_{oc}$, J$_{sc}$, or solar conversion efficiency. This method is expected to have a greater effect when this Na$^+$ ion-diffusion barrier is combined with the insertion of another barrier layer between the anti-reflection coatings and solar cells.

**Author Contributions:** Conceptualization, E.J., K.-s.O. and S.R.; methodology, E.J.; experimental design, sample fabrication, data curation, E.J. and K.-s.O.; writing, E.J.; review and editing, S.R.; supervision, project administration, funding acquisition, S.R. All authors have read and agreed to the published version of the manuscript.

**Funding:** This work was supported by the Korea Institute of Energy Technology Evaluation and Planning (KETEP) and the Ministry of Trade, Industry & Energy (MOTIE) of the Republic of Korea (No. 20183010014320). This work was also supported by Korea Institute for Advancement of Technology (KIAT) grant funded by the Korea Government (MOTIE) (P0017012, Human Resource Development Program for Industrial Innovation). Also, this work was supported by Kyonggi University's Graduate Research Assistantship 2020.

**Data Availability Statement:** Not applicable.

**Acknowledgments:** We would like to thank Soo Min Kim at the Gumi Electronics & Information Technology Research Institute regarding the formation of module-like layer structures and the accelerated PID test.

**Conflicts of Interest:** The authors declare no conflict of interest.

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
