# Peer review of "Reduction of Potential-Induced-Degradation of p-Type PERC Solar Cell Modules by an Ion-Diffusion Barrier Layer Underneath the Front Glass"

_processes, doi:10.3390/pr10020334_

Round 1

Reviewer 1 Report

Review Comments on the Manuscript processes-1577570

This paper is very interesting. In the paper, the authors explored the influence of Al2O3 thin layers on the reduction of potential-induced-degradation (PID) in silicon solar cell modules. This layer is known in the literature as high light transmittance with outstanding barrier performance against ion diffusion. During investigation was applied 30 nm layer of Al2O3 was, which was deposited by the ALD method. Its application aimed to suppress PID effectively.

  1. Editing

  1. Line 23 - There is “serves and high energy conversion efficiency. [1,2]”, but should be” serves and high energy conversion efficiency [1,2].” The period should be outside the quotation.
  2. Line 33 - there should be a comma before "because".
  3. Lines: 35, 38, 47, 62, 64, 68,70 - The period should follow the citation.
  4. Line 53 - There is “current, [12-18] which”, but should be “current [12-18], which”.
  5. Line 70 - Literature citations should be consecutive.
  6. Lines: 105-107 - The signature must be directly under the figure 1.Text from lines 102-104 should be moved higher.
  7. Line 132 – there is “AleO3”, but sholud be ’’Al2O3’’.
  8. Review the literature again and refer to it for subscripts and superscripts, e.g. in item 7. There is ’’ 26th Eur.”, but should be ’’ 26th ”.

  1. Substantive

  1. The paper lacks detailed information on the mini-modules used. How many solar cells did the module consist of? What was their size, thickness, and so of?
  2. Were other Al2O3 layer thicknesses also applied using the ALD method?
  3. There are no references to literature from the last two years in the paper.

Therefore, I recommend that this manuscript consider publication, after taking into account the editing and substantive corrections.

Author Response

We thank the reviewer for the meticulous review. Our answer is given in blue for the editing issue (A) and substantive issue (B) raised by the reviewer. The manuscript has been revised accordingly.  

A) Editing

  1. Line 23 – There is “serves and high energy conversion efficiency. [1,2]”, but should be “serves and high energy conversion efficiency [1,2].” The period should be outside the quotation.
  2. Line 33 - there should be a comma before “because”.
  3. Lines: 35, 38, 47, 62, 64, 68,70 - The period should follow the citation.
  4. Line 53 - There is “current, [12-18] which”, but should be “current [12-18], which”.
  5. Line 70 - Literature citations should be consecutive.
  6. Lines: 105-107 - The signature must be directly under the figure 1. Text from lines 102-104 should be moved higher.
  7. Line 132 there is AleO3 , but sholud be Al2O3 .
  8. Review the literature again and refer to it for subscripts and superscripts, e.g. in item 7. There is 26th Eur. , but should be 26th Eur. .

Regarding #1, #3, #4, we double checked the format for the reference notation in the text and revised all the reference numbers to be placed before the punctuation. Regarding #2, we added a comma in the right place. Regarding #5, we put the corresponding reference at [25]. Regarding #6, we revised the position of the figure so that the figure and caption should be seen appropriately. Regarding #7, the typo was corrected. Regarding #8, the superscript was revised. 

B) Substantive

  1. The paper lacks detailed information on the mini-modules used. How many solar cells did the module consist of? What was their size, thickness, and so of?

There might be misunderstanding regarding on the term “mini-module”. As S. Yamaguchi and K. Ohdaira reported in their paper (Solar Energy, Vol. 155, pp.739-744 (2017)), the cell-level accelerated PID test utilized in this work does not require the real modules but module-like layer stacks without lamination as depicted in Figure 1. In this case, each 5 x 5 cm2 sample corresponds to one small module and the 2-kg weight gives the effect of the lamination. We revised the relevant terminology in the manuscript.

  1. Were other Al2O3 layer thicknesses also applied using the ALD method?

As it is presented in Figure 4, the reduction of cell efficiency loss was almost saturated, and the defect states analyzed by the electroluminescence was almost same for the sample with 30 nm-thick Al2O3. Therefore, we expected there would not be any significant change beyond 30 nm Al2O3.

  1. There are no references to literature from the last two years in the paper.

We added two references, ref. #5 and #12, from the last two years. 

Reviewer 2 Report

I have read your manuscript, however, it has not been accepted for the current version. I have a few points as below;

1) Apart from the EL image, other analyses (SIMS) are needed to prove an ion diffusion barrier in the module. Why the cell efficiency was degraded without and with ion diffusion barrier. 

2) In fig 4 (b), what is the red-color line? Is it a fitting line or not? I think that no need for the red-color line and even fig 4 (b) also no need. Please take this into consideration.  

I think that you will be referred to do it; 
H. Park et al., Microelectronic Engineering 216, 111081 (2019), A reliability study of silicon heterojunction photovoltaic modules exposed to damp heat testing

Author Response

I have read your manuscript, however, it has not been accepted for the current version. I have a few points as below;

1) Apart from the EL image, other analyses (SIMS) are needed to prove an ion diffusion barrier in the module. Why the cell efficiency was degraded without and with ion diffusion barrier. 

2) In fig 4 (b), what is the red-color line? Is it a fitting line or not? I think that no need for the red-color line and even fig 4 (b) also no need. Please take this into consideration.  

I think that you will be referred to do it; 
H. Park et al., Microelectronic Engineering 216, 111081 (2019), A reliability study of silicon heterojunction photovoltaic modules exposed to damp heat testing

We thank the reviewer for the review. Our answer is given in blue for each question. The manuscript has been revised accordingly.

1) Unfortunately, the cells were damaged during the disassembly for the subsequent analysis like SIMS that the reviewer suggested. We would like to add a reference that analyzed the regions where potential-induced-degradation (PID) occurred using electroluminescence (EL), lock-in thermography (LIT), time-of-flight secondary ion mass spectroscopy (ToF-SIMS). As shown in the reference [V. Naumann et al., Energy Procedia 27, pp.1-6 (2012)], an accumulation of alkali metals at the interface of the front side coatings of the solar cell was found at the degraded regions. From these results and our EL data, it can be implied that 30 nm-thick Al2O3 can suppress the migration of Na+ ions effectively preventing PID. We added this explanation as well as the reference above in the manuscript.

2) The red line is a fitting curve that was expected for different thickness of Al2O3. We agree to the reviewer, and we deleted the red line. Also, we added the reference that the reviewer suggested.

In addition, we added the reference [Microelectronics Engineering 216, 111081 (2019)] that the reviewer mentioned about the damp heat PID test.
